# Assessment of Fresh Miscanthus Straw as Growing Media Amendment in Nursery Production of *Sedum spectabile* ‘Stardust’ and *Hydrangea arborescens* ‘Annabelle’

**DOI:** 10.3390/plants12081639

**Published:** 2023-04-13

**Authors:** Magdalena Pancerz, Marta Czaplicka, Przemysław Bąbelewski

**Affiliations:** 1Department of Horticulture, Faculty of Life Sciences and Technology, Wroclaw University of Environmental and Life Sciences, Grunwaldzki Sq. 24a, 50-363 Wroclaw, Poland; 2Department of Food, Agricultural and Biological Engineering, The Ohio State University, 1680 Madison Ave, Wooster, OH 44691, USA

**Keywords:** soilless substrates, peatmoss, ornamental plants, fertilization

## Abstract

The aim of this research was to assess the influence of fresh miscanthus straw shreds as a component of growing media in nursery production of perennial *Sedum spectabile* ‘Stardust’ and woody shrub *Hydrangea arborescens* ‘Annabelle’. A total of five substrate mixes composed of peatmoss and miscanthus straw were used: 100%P, 70%P:30%M, 50%P:50%M, 70%P:30%M, 100%M. Each substrate was subjected to three fertilizer treatments: Basacote, Basacote + YaraMila, and YaraMila. The growth response of both tested species was very similar. In general, plants performed best in 100%P, and the quality decreased with increasing miscanthus straw amendment; however, differences in height and dry weight at the level of ~9% suggest that *Sedum* plants obtained market value with up to 50% miscanthus amendment and *Hydrangea* plants with up to 30% miscanthus mixed in media. The most favorable effect on the tested parameters was a combination of Basacote + YaraMila, which delivered more soluble salts, and in higher rates than Basacote and YaraMila used separately. Decrease in EC and nutrients in the substrate with increase in miscanthus straw amendment suggest that uniform irrigation among all the treatments contributed to nutrients leaching from miscanthus media due to its lower water holding capacity.

## 1. Introduction

Greenhouse and nursery production depends on peatmoss as the basic growing medium. Increasing plant production requires progressively greater use of soilless substrates in the horticultural industry. Block et al. [1] predicts that global demand for soilless substrates will double in the next 20 years. Despite the many advantages of peatmoss, environmental concerns related to peat harvesting [2,3] have led to the view of peatmoss as an unrenewable and unsustainable material [4] and, furthermore, to the establishment of legal protection of wetlands in some areas of the world [5]. Reduced availability and uncertainty regarding peatmoss in both the near and distant future have opened investigation of materials that could substitute for peat in soilless substrates.

The search for new materials suitable for container production is occurring in two ways: finding components that reduce the quantity of peatmoss used in the mix, or completely replacing peatmoss. Two raw materials that were profitably and successfully used as peatmoss substitutes are pine bark and coir, used separately or in mixes [6,7,8,9]. Pine bark has been used as a soilless medium alongside peatmoss for several decades [10,11,12,13] and is being subsequently replaced by wood materials [14,15,16,17]. However, changes in the forest industry are causing a decrease in wood derived materials available for horticultural production and are modifying the dynamics relating to woody materials obtainable for soilless media. Coir, as a waste material from coconut husks, was first used in tropical countries as a locally available material for soilless substrates [18]. With the development of processing and compressing it into bricks or bales, allowing for easier transportation [19,20], it became a peat replacement in container production in other areas of the world [21,22,23]. However, ecological concerns related to the carbon footprint associated with coir transportation, as well as the erratic quality of the raw material, very often showing high salinity and other contaminations affecting plant growth [22,24], are slowly sidetracking this material when considering it as peatmoss substitute.

As highlighted by [25], to achieve an environmentally sustainable substrate, raw materials have to be selected with regard to their performance under practical conditions, economical aspects, including equal or better characteristics than commercially available growing media, and ecological aspects that consider the sustainability of any given material. Among renewable resources that should be considered as raw materials for soilless substrates, energy crops come into play due to their fast and high biomass production that has relatively low cultivation requirements compared to other crops. Miscanthus (*Miscanthus × giganteus*), a tall hybrid C4 grass genotype with proven utility as a biomass crop, was described by Heaton et al. [26] as one of the most productive land plants in temperate climates. This statement was confirmed by Dohleman and Long [27], who showed in their studies that miscanthus was more productive and efficient than maize and other C4 cols resistant biomass crops. Moreover, miscanthus plants, after a few years of establishment, are characterized by stable biomass and adequate biomass quantity [28]. This led several researchers to conduct trials into the use of miscanthus straw as potting medium [29,30,31,32,33,34].

As there are few studies on the use of miscanthus straw in ornamental plant production, the authors decided to assess plant performance in miscanthus straw amended media on two selected ornamental species: perennial *Sedum spectabile* ‘Stardust’ and woody shrub *Hydrangea arborescens* ‘Annabelle’, followed by the chemical analysis of leaf tissues of the tested species and substrate analyses.

## 2. Results and Discussion

### 2.1. Plant Performance Measurements

#### 2.1.1. Height, Diameter and Shoot Growth

Both *Sedum* (Table 1) and *Hydrangea* (Table 2) were the tallest with the widest diameter when grown in 100% peatmoss, and the shortest with the smallest diameter when grown in miscanthus straw. For *Hydrangea,* the difference in height compared with miscanthus was about 50% less when compared to the peatmoss control, and for *Sedum* a difference of around 50% was noticed in diameter. In general, both species decreased in height and diameter with the increase of miscanthus straw in the media, which suggests that water availability for plants decreased along with increasing miscanthus amendment in the substrate. Similar results were obtained by Tsakaldimi and Ganatas [35] growing seedlings of three native tree species, where the use of kenaf in the substrate resulted in significant reduction in seedling dimensions, height, and diameter, with an increased proportion of rice hulls. Starr et al. [36], growing seedlings of bald cypress, Chinese pistachio and silver maple in substrates with the amendment of eastern redcedar, noticed that less growth occurred when plants were grown in 80% amendment of eastern redcedar in media, while *Sedum* and *Hydrangea* mixes containing 70% and 100% seemed to cause the biggest decrease the height and diameter of plants. *Sedum* and *Hydrangea* were the tallest and with a larger diameter when fertilized with Basacote + YaraMila. These results indicate that, for plants with high nutritional requirements, using controlled released fertilizer together with easy soluble fertilizer can be the most beneficial practice. Both *Sedum* and *Hydrangea* were the shortest and with the smallest diameter in miscanthus straw with Basacote fertilizer. This is most likely linked to the lower water holding capacity of miscanthus straw in comparison to peatmoss media, resulting in excessive leaching of nutrients, or to not enough provision of saturation to release nutrients from the coated slow release fertilizer. The highest *Sedum* plants that were the largest in diameter were found in peatmoss with Basacote + YaraMila fertilizer, and foe *Hydrangea* in 100% peatmoss and 70% peatmoss media with YaraMila. In terms of main shoot number and length, *Hydrangea* expressed similar tendencies to those for height and diameter, having the greatest number of the longest shoots in 100%P substrate and in Basacote + YaraMila fertilizer (Table 2). As noticed by Roosta and Afsharipoor [37], vigorous vegetative growth observed in peat is most likely due to higher nutrient uptake and possibly the high water content.

#### 2.1.2. Flowering

*Sedum* produced the greatest inflorescence number (Table 3), and was also the tallest, in peatmoss substrate, and was the least, shortest, and smallest in diameter in 100% miscanthus. This suggests that nutrients, especially the phosphorus responsible for flowering, was leaching in much greater amounts from miscanthus based media, negatively affecting the generative stage of plants. Harris et al. [38] observed that petunia at the end of the production phase had the lowest number of flowers in peat:wood and peat:fiber media, in comparison to peat:coir mixes. Awang et al. [39] also noted a negative effect of media containing kenaf core fiber on flower size of *Celosia cristata.* For *Sedum,* the most favorable for flower features was fertilization with Basacote + YaraMila, delivering more soluble salts from the two different types of fertilizer. Among the substrate x fertilizer interactions, the least favorable was miscanthus substrate with Basacote, which can be affected by the low water holding capacity of miscanthus straw that did not allow Basacote, as a coated fertilizer, to release nutrients to the medium. As highlighted by Niemiera and Leda [40], N leaching losses in liquid fertilizer are higher than in controlled released fertilizer; however, depending on the application rate, relative N losses from CRF can be significant and reach 12–23%. These data can suggest that phosphorus, which has high mobility in soilless media and is prone to leaching, could also be subject to similar tendencies as with nitrogen.

#### 2.1.3. Leaf Measurements

Both *Sedum* (Table 4) and *Hydrangea* (Table 5) showed similar leaf response to growing media mixes: leaf number, length, width and leaf blade area of both species tended to be the greatest in plants grown in peatmoss and the lowest in those cultivated in miscanthus straw. Similar results were observed by Bassan at al. [41] in leaf number and area of tomato transplants, negatively affected by increasing rates of rice hull in the media. As both miscanthus and rice hulls have lower water holding capacity than peatmoss, these results can be affected by water availability. As Saberi et al. [42] noticed in forage sorghum, leaf area of plants was reduced in response to decreasing water availability. *Hydrangea* produced the lowest number of the smallest leaves in Basacote, and Prince et al. [43] similarly noticed that one time application of a standard rate of controlled released fertilizer had a negative effect on leaf area of potted chrysanthemum. *Sedum* had the lowest number of the smallest leaves in YaraMila. Both species had the lowest number of the smallest leaves when grown in miscanthus with the use of Basacote. For *Sedum,* the most favorable for leaf growth and development were media containing 100%, 70% and 50% peatmoss, with the use of Basacote + YaraMila, while for *Hydrangea,* peatmoss with YaraMila fertilization was best.

#### 2.1.4. Dry Biomass

*Sedum* accumulated the highest dry weight of roots in peatmoss media, and the highest shoot dry weight in 70% peatmoss (Table 6). Gomez and Robbins [44] found that shoot dry weight of spirea was significantly greater with up to 40% of rice hulls in the media, and decreased as the percentage of rice hulls increased in the blends. *Hydrangea* had the highest fresh weights of shoot part and roots in 70% peatmoss, but the highest dry weight for both was in 100% peatmoss (Table 7). Tsakaldimi and Ganatas [35], growing seedlings of three native tree species, observed that the use of kenaf decreased seedling biomass, what confirms the reaction of *Sedum,* which produced the lowest fresh and dry weight of shoot parts and roots when grown in 100% miscanthus. In general, the biomass of both species decreased with the increase of miscanthus straw in the media. Similarly, Webber et al. [45] found that shoot and root dry weights of *Vinca minor* decreased as the percentage of kenaf increased. Frangi et al. [46] obtained similar results, where shoot dry weight of cherry laurel was negatively influenced when the quantities of *Arundo donax* and *Miscanthus sinensis* fiber increased in the growing medium. In both species, the highest dry weights were noticed in Basacote + YaraMila fertilizer, and plants grown only in YaraMila were characterized by the lowest fresh and dry weights of shoot parts and roots. *Hydrangea* showed a highly unified response on substrate × fertilizer treatment and had the highest dry weights in 70% peatmoss with Basacote + YaraMila and the lowest in miscanthus fertilized with YaraMila.

### 2.2. Analyses of Leaves

Higher concentration of chlorophylls in *Sedum* (Table 8) were in general found in media containing 70% and 100% miscanthus, while in *Hydrangea* (Table 9) all chlorophylls were highest in peatmoss, which supports the color reading of the leaves when regarding the darkest leaves with the bluest tone in this medium (data not shown).

Chlorophyll contents in *Hydrangea* and *Sedum* were the highest in Basacote, with YaraMila species having the darkest leaves with the greenest tone and the bluest tone in this fertilization treatment.

The lowest chlorophyll content in *Sedum* was in YaraMila fertilization, while for Hydrangea this was in *Basacote.* For *Sedum,* there was no clear pattern of chlorophyll concentrations within substrate × fertilizer treatment. On the other hand, in *Hydrangea*, all tested chlorophylls were the highest in peatmoss with Basacote + YaraMila, and lowest, as for leaf color readings, in miscanthus substrate with YaraMila fertilizer. Contradictory findings were shown in strawberry cultivation in coir, peat, reed canary grass and a mix of peat with reed canary grass, where strawberry leaf chlorophyll content did not differ between the treatments [47].

Foliar nutrients did not show any tendencies and their highest and lowest values were spread among substrate types within the species, and there were no consistent tendencies when comparing nutritional status of leaves of both *Sedum* (Table 10) and *Hydrangea* (Table 11). A similar situation occurred in fertilizer and in substrate × fertilizer treatments. As suggested by Mustafa et al. [48], different media amendments can display different extractability and, due to many interactions between media components, cause discrepancies between substrate and foliar nutrient levels.

### 2.3. Substrate Analyses

In substrates of both *Sedum* (Table 12) and *Hydrangea* (Table 13), pH and EC were the highest in peatmoss media, and the lowest in miscanthus. This suggests that both species were taking up and/or leaching soluble salts from miscanthus media much more quickly, reducing its EC. Additionally, as Altland [49] found, in general substrates made from bioenergy crops have pH values higher than recommended. In both species, pH tends to be higher in Basacote fertilizer and lower in Basacote + YaraMila, while EC was highest in Basacote + YaraMila and lowest in YaraMila. Such high EC values in both *Sedum* and *Hydrangea* can be affected by soluble salts uptake in these species. Hicklenton and Cairns [50] noticed that the EC of juniper container leachate was higher than in cotoneaster and suggested that available nutrients were not absorbed as readily by juniper.

Nutrients in substrates of *Sedum* (Table 14) tend to show the highest contents in 100% and 70% peatmoss media, while in *Hydrangea* (Table 15) they were spread among the substrates. In *Sedum,* the substrate × fertilizer treatment that affected the highest N, nitrates and K was 70% peatmoss with Basacote + YaraMila, and for Ca and Mg 50% peatmoss with YaraMila. *Hydrangea* had the highest P level in miscanthus straw. Frangi et al. [46] noted that P content increased when *Arundo donax* and *Miscanthus sinensis* rate in the medium was higher. However, *Sedum* in our own research did not fall into this pattern and had the highest P content in 100% peatmoss. As mentioned by Evans et al. [51], nutrients in rice hulls and other compost amendments can display varying degrees of extractability when mixed in the media, due to complex interactions between media components. Furthermore, these interactions can explain some of the discrepancies between nutrients measured in different blends [48].

## 3. Materials and Methods

The research study was conducted at the Research Development Station of Wroclaw University of Environmental and Life Sciences. Plant trial was established in mid-May of 2014 and repeated for two consecutive years. A 5 × 3 factorial experiment was arranged in a randomized block design with a total of 15 treatments consisting of 24 plants each (eight plants in three replications).

### 3.1. Plant Material and Treatments

For the purpose of this research study, two plant species from different groups were selected: perennial *Sedum spectabile* ‘Stardust’ and woody shrub *Hydrangea arborescens* ‘Annabelle’. Both species are commonly grown ornamental plants with high nutritional requirements. Plants were propagated at the Research Development Station of Wroclaw University of Environmental and Life Sciences in plugs and used in the form of rooted cuttings as the starting material for this study.

The two main factors in the research were substrate mix and fertilizer type.

The first factor was a substrate mixture composed of different proportions of peatmoss and shredded miscanthus straw:100% peatmoss (control)70% peatmoss + 30% miscanthus50% peatmoss + 50% miscanthus30% peatmoss + 70% miscanthus100% miscanthus

The second factor was the two different fertilizer types: controlled released fertilizer Basacote (Basacote^®^ Plus 6M 16-8-12(+2+TE); Compo) and water soluble YaraMila Complex (12 N(5 N-NO_3_+7 N-NH_4_+11 P_2_O_5_+18 K_2_O+2.7 MgO+20 SO_3_+0.015 B+0.2 Fe+0.02 Mn+0.02 Zn; Yara), used separately and in a mix with different fertilization schemes:3 g.dm^−3^ of Basacote premixed with each substrate mix3 g.dm^−3^ of Basacote premixed with each substrate mix with YaraMila Complex top dressing, 3 times during vegetation period. at a dose of 1 g.dm^−3^1 g.dm^−3^ of YaraMila Complex premixed with each substrate mix with YaraMila Complex top dressing, 3 times during vegetation period, at a dose of 1 g.dm^−3^

Fresh miscanthus straw (*Miscanthus* × *giganteus* Greef et Deu) was delivered from the experimental station of the Wroclaw University of Environmental and Life Sciences in Pawlowice, shredded in a hammermill and then screened to a particle size not exceeding 4 × 2 × 0.5 cm. To decrease the high carbon to nitrogen ratio in miscanthus close to the optimal level (24:1), shredded straw was premixed with YaraMila Complex (rate calculated based on N content of fertilizer and C:N ratio in starting material; data not shown). Peatmoss (sphagnum peatmoss, Klasmann) was mixed with miscanthus straw in proper ratios, samples of each substrate were taken to determine pH (Elmetron (CPI-501)), and then amended with the proper amount of lime to establish pH at the level of 6.2–6.5 based on the neutralization curve (data not shown). All five media were split into three piles and mixed with fertilizers: Basacote, Basacote with YaraMila, and YaraMila. Rooted cuttings of *Aster* and *Spiraea* were transplanted into 3 L pots filled with the proper substrate × fertilizer mix and placed in the outdoor nursery on black nursery fabric. Plants were trimmed by 1/3 height to stimulate shoot growth, and pots were fully saturated with water. Watering continued throughout the entire vegetation period until plants were ready to be measured. Irrigation was performed using overhead irrigation as needed, on average 3 times a week, with 300 mL of water per 1 dm^−3^ of substrate.

### 3.2. Plant Performance Measurements

For *Sedum*, measurements were taken in its full flowering stage, and for *Hydrangea*, by the end of the vegetation period. Biometric measurements for both species included:plant height (measured from the level of substrate to the highest shoot)plant diameter (measured at the widest and narrowest axis and averaged)

Additional *Hydrangea* measurements included:main shoot numbermain shoot length (measured for all main shoots and averaged)

Leaf performance of both species was tested on 8 randomly selected plants within the treatment and included:leaf number (total leaves on both main and side shoots)leaf blade lengthleaf blade widthleaf blade area

Leaf blade length, width and area were tested on 15 leaves per plant, selecting mature, fully developed leaves in the central area of the plant, using a field portable leaf meter AM 300 (Opti-Sciences Inc., Hudson, NH, USA)

Additionally, for *Sedum*, the inflorescence number per plant, average inflorescence height and diameter were assessed. *Hydrangea* did not flower in the first year after planting.

Fresh shoot and root weight for both species was measured immediately after performing the above mentioned measurements. Dry weights were measured after drying above ground, and root biomass in the oven at 70 °C for 72 h.

### 3.3. Leaf Analyses

From each of the two tested species 20–25 leaves per plants within the substrate × fertilizer treatment (total of 8 randomly selected plants within each treatment) were collected and unified. Three subsamples from each treatment were subjected to testing:in fresh material, chlorophyll a,b and total using spectrophotometry by Arnon [52]

leaf brightness and two color tones, using HunterLab MiniScan EZ working in CIE L * a * b * scale that was recalculated to RGB scale using color converter https://www.nixsensor.com/free-color-converter/ (accessed on 13 February 2023). Fresh leaf blades were spread flat on a white sheet, a reading head was placed on the leaf blade ensuring that the leaf surface covered the reading area, and measurements were taken and downloaded from the device

Chemical analyses of leaves included:P and Mg by the colorimetric method of King [53] (Spectrophotometer S106 WPA),K and Ca by flame photometry as in Toth and Prince [54] (Carl Zeiss Jena flame photometer),NO_3−_ by flow colorimetry by Shinn [55].

### 3.4. Substrate Analyses

To perform chemical analyses of tested substrate mixes, 500 mL of substrate from 8 randomly selected plants within the substrate x fertilizer treatment of each species was collected and unified. Three subsamples from each treatment were subjected to testing: electrical conductivity (EC) measurements were made with a conductivity meter (Orion model 142) and pH of the soil with Elmetron (CPI-501) at a soil:distilled water ratio of 1:2. Total N was measured by the Kjedahl method, P and Mg by the colorimetric method (Spectrophotometer S106 WPA), K and Ca by flame photometry (Carl Zeiss Jena flame photometer), and NO_3−_ by flow colorimetry.

### 3.5. Statistical Analysis

The data were subjected to analysis of variance (ANOVA). The *F*-test was used to identify the treatments’ main effects and interactions, followed by Fisher’s range test at the 0.05 significance level using Statistica 13.3.721.0. As the research was conducted over three consecutive years and statistical analyses did not show significant differences between years, averaged data from 2014, 2015 and 2016 were analyzed and shown together.

## 4. Conclusions

Both *Sedum* and *Hydrangea* performed best in 100%P media and plant height decreased along with increase in miscanthus amendment. However, *Sedum* height in 50% miscanthus amended media was ~9% lower than in peatmoss and, for *Hydrangea* with 30% miscanthus amendment, this was ~9% lower than in peatmoss. A similar tendency was observed in both species with dry weights. Based on these results, it can be considered that, for *Sedum,* miscanthus amendment up to 50% and, for *Hydrangea,* up to 30% still produced plants with market value. EC and nutrient content in substrates decreased along with increase in miscanthus straw amendment in media. The lower water holding capacity of miscanthus straw suggests that the same irrigation frequency and amount could cause nutrient leaching, and further investigation is needed to develop more suitable practices for this media component.

## Figures and Tables

**Table 1 plants-12-01639-t001:** Selected biometric features of *Sedum spectabile* ‘Stardust’ grown in containers with five different substrates.

Substrate (A)	Fertilization (B)			
	Basacote^®^ Plus 6M 16-8-12(+2 + TE) 2 doses	Basacote + YaraMila	YaraMila 12N(5N-NO_3_+7N-NH_4_) +11P_2_O_5_+18K_2_O +2.7MgO +20SO_3_ +0.015B+0.2Fe +0.02Mn +0.02Zn	Mean A
	Height (cm)			
100% P	30.7 e	37.6 a	34.4 bc	34.2 a
70%P + 30%M	32.9 d	35.5 b	28.9 f	32.4 b
50%P + 50%M	30.7 e	33.8 cd	26.3 g	30.3 c
30%P + 70%M	27.3 g	39.9 ef	26.2 gh	27.8 d
100%M	23.0 i	28.7 f	24.9 h	25.5 e
Mean B	28.9 b (32.9–23.0)	33.1 a (39.9–28.7)	28.1 c (34.4–24.9)	
	Diameter (cm)			
100% P	25.9 c	29.4 a	27.9 b	27.7 a
70%P + 30%M	24.2 e	26.4 c	21.2 f	23.9 b
50%P + 50%M	21.2 fg	25.0 d	15.1 j	20.4 c
30%P + 70%M	17.3 i	20.5 gh	14.7 j	17.5 d
100%M	13.0 k	19.9 h	15.0 j	16.0 e
Mean B	20.3 b (25.9–13.0)	24.24 a (29.4–19.9)	18.8 c (27.9–14.7)	

(A) composed of various combinations of peatmoss (P) and miscanthus straw (M), along with one of three fertilizers (B), including either Basacote (15-11-13) or water soluble YaraMila Complex (12-5-15), either alone or in combination. Different lower case letters within mean A, mean B and A × B interaction indicate statistically significant differences at the significance level of (*p* < 0.05) by Fisher’s test.

**Table 2 plants-12-01639-t002:** Selected biometric features of *Hydrangea arborescence* ‘Annabelle’ grown in containers with five different substrates.

Substrate (A)	Fertilization (B)			
	Basacote	Basacote + YaraMila	YaraMila	Mean A
	Height (cm)			
100% P	43.5 c	47.1 b	51.2 a	47.3 a
70%P + 30%M	37.0 e	49.7 a	40.8 d	42.5 b
50%P + 50%M	34.6 f	44.2 c	30.8 g	36.6 c
30%P + 70%M	30.9 g	29.1 h	25.5 h	28.5 d
100%M	20.7 j	21.4 j	23.2 i	21.8 e
Mean B	33.3 c (43.5–20.7)	38.3 a (49.7–21.4)	34.3 b (51.2–23.2)	
	Diameter (cm)			
100% P	34.2 c	38.1 b	40.7 a	37.7 a
70%P + 30%M	30.8 d	40.1 a	30.7 d	33.9 b
50%P + 50%M	34.8 c	40.7 a	26.6 f	33.7 b
30%P + 70%M	26.2 f	28.2 e	27.5 ef	27.3 c
100%M	16.2 i	19.9 h	23.5 g	19.9 d
Mean B	28.3 c (34.8–16.2)	33.4 a (40.7–19.9)	29.8 b (40.7–23.5)	
	Main shoot number			
100% P	3.8 cd	4.3 b	3.9 c	3.9 a
70%P + 30%M	3.6 d	3.1 e	2.4 g	3.0 c
50%P + 50%M	2.8 f	4.9 a	2.4 g	3.3 b
30%P + 70%M	2.4 g	2.0 h	3.2 e	2.5 d
100%M	1.7 i	2.4 g	1.5 i	1.8 e
Mean B	2.8 b (3.8–1.7)	3.3 a (4.3–2.0)	2.7 c (3.9–1.5)	
	Main shoot length (cm)			
100% P	32.5 d	36.8 b	41.6 a	36.9 a
70%P + 30%M	29.5 ef	41.3 a	34.5 cd	34.8 b
50%P + 50%M	28.5 f	34.6 c	30.7 e	31.3 c
30%P + 70%M	29.0 ef	25.7 g	25.6 g	26.8 d
100%M	16.0 i	18.8 h	19.6 h	18.1 e
Mean B	27.1 c (32.5–16.0)	31.5 a (41.3–18.8)	30.2 b (41.6–19.6)	

(A) composed of various combinations of peatmoss (P) and miscanthus straw (M), along with one of three fertilizers (B), including either Basacote (15-11-13) or water soluble YaraMila Complex (12-5-15), either alone or in combination. Different lower case letters within mean A, mean B and A × B interaction indicate statistically significant differences at the significance level of (*p* < 0.05) by Fisher’s test.

**Table 3 plants-12-01639-t003:** Flowering of *Sedum spectabile* ‘Stardust’ grown in containers with five different substrates.

Substrate (A)	Fertilization (B)			
	Basacote	Basacote + YaraMila	YaraMila	Mean A
	Inflorescence Number			
100% P	6.7 ab	6.6 c	6.8 ab	6.7 a
70%P + 30%M	4.9 gh	6.7 ab	6.1 e	5.9 c
50%P + 50%M	6.0 de	7.0 a	5.1 fg	6.1 b
30%P + 70%M	4.1 i	5.2 f	4.7 h	4.7 d
100%M	4.0 i	5.2 f	5.1 fg	4.7 d
Mean B	5.2 c (6.7–4.0)	6.09 a (7.0–5.2)	5.6 b (6.8–4.7)	

(A) composed of various combinations of peatmoss (P) and miscanthus straw (M), along with one of three fertilizers (B), including either Basacote (15-11-13) or water soluble YaraMila Complex (12-5-15), either alone or in combination. Different lower case letters within mean A, mean B and A × B interaction indicate statistically significant differences at the significance level of (*p* < 0.05) by Fisher’s test.

**Table 4 plants-12-01639-t004:** Selected leaf features of *Sedum spectabile* ‘Stardust’ grown in containers with five different substrates.

Substrate (A)	Fertilization (B)			
	Basacote	Basacote + YaraMila	YaraMila	Mean A
	Leaves Number			
100% P	84.8 a	70.2 b	50.4 c	68.5 a
70%P + 30%M	40.6 e	40.4 e	43.2 d	41.4 b
50%P + 50%M	39.5 e	41.1 e	26.3 g	35.6 c
30%P + 70%M	27.8 g	30.0 f	24.4 h	27.4 d
100%M	20.5 j	23.3 hi	21.7 ij	21.8 e
Mean B	42.6 a	41.0 b	33.2 c	
	Leaf blade area (cm^3^)			
100% P	14.29 e	19.56 a	17.61 b	17.15 a
70%P + 30%M	17.04 c	17.26 bc	16.13 d	16.81 b
50%P + 50%M	14.23 e	19.38 a	9.33 h	14.31 c
30%P + 70%M	11.22 g	15.83 d	9.62 h	12.22 d
100%M	6.52 j	12.84 f	8.11 i	9.16 e
Mean B	12.66 b	16.98 a	12.16 c	

(A) composed of various combinations of peatmoss (P) and miscanthus straw (M), along with one of three fertilizers (B), including either Basacote (15-11-13) or water soluble YaraMila Complex (12-5-15), either alone or in combination. Different lower case letters within mean A, mean B and A × B interaction indicate statistically significant differences at the significance level of (*p* < 0.05) by Fisher’s test.

**Table 5 plants-12-01639-t005:** Selected leaf features of *Hydrangea arborescence* ‘Annabelle’ grown in containers with five different substrates.

Substrate (A)	Fertilization (B)			
	Basacote	Basacote + YaraMila	YaraMila	Mean A
	Leaves Number (Total)			
100% P	59.8 d	68.8 c	71.3 b	66.6 a
70%P + 30%M	60.1 d	55.7 e	50.4 f	55.4 b
50%P + 50%M	44.6 h	85.1 a	34.5 j	54.7 b
30%P + 70%M	29.9 k	36.7 i	23.8 m	30.2 d
100%M	34.1 j	26.0 l	46.9 g	35.7 c
Mean B	45.7 b	54.5 a	45.4 b	
	Leaf blade area (cm^3^)			
100% P	50.69 e	63.73 c	84.50 a	66.31 a
70%P + 30%M	63.05 c	69.39 b	58.08 d	63.51 a
50%P + 50%M	39.40	53.98 de	47.33 ef	46.90 b
30%P + 70%M	45.80 f	43.35 f	41.67 f	43.61 b
100%M	15.82 h	27.64 g	26.56 g	23.34 c
Mean B	42.95 b	32.79 c	51.63 a	

(A) composed of various combinations of peatmoss (P) and miscanthus straw (M), along with one of three fertilizers (B), including either Basacote (15-11-13) or water soluble YaraMila Complex (12-5-15), either alone or in combination. Different lower case letters within mean A, mean B and A × B interaction indicate statistically significant differences at the significance level of (*p* < 0.05) by Fisher’s test.

**Table 6 plants-12-01639-t006:** Fresh and dry weight of *Sedum spectabile* ‘Stardust’ grown in containers with five different substrates.

Substrate (A)	Fertilization (B)			
	Basacote	Basacote + YaraMila	YaraMila	Mean A
	Shoot Fresh Weight (g)			
	Shoot dry weight (g)			
100% P	115.22 e	166.37 c	137.49 d	139.69 b
70%P + 30%M	132.91 d	200.51 a	102.32 f	145.25a
50%P + 50%M	131.51 d	193.70 b	54.30 h	126.50 c
30%P + 70%M	99.32 f	113.70 e	73.19 g	95.40 d
100%M	45.18 i	118.53 e	75.65 g	79.79 e
Mean B	104.83 b	158.56 a	88.59 c	
	Root dry weight (g)			
100% P	30.26 b	28.20 c	23.44 fg	27.30 a
70%P + 30%M	25.70 e	24.04 f	19.91 i	23.22 c
50%P + 50%M	27.35 d	33.63 a	17.59 j	26.19 b
30%P + 70%M	22.57 h	24.06 f	17.33 j	21.32 d
100%M	14.36 k	22.83 gh	16.96 j	18.05 e
Mean B	24.05 b	26.55 a	19.05 c	

(A) composed of various combinations of peatmoss (P) and miscanthus straw (M), along with one of three fertilizers (B), including either Basacote (15-11-13) or water soluble YaraMila Complex (12-5-15), either alone or in combination. Different lower case letters within mean A, mean B and A × B interaction indicate statistically significant differences at the significance level of (*p* < 0.05) by Fisher’s test.

**Table 7 plants-12-01639-t007:** Fresh and dry weight of *Hydrangea arborescence* ‘Annabelle’ grown in containers with five different substrates.

Substrate (A)	Fertilization (B)			
	Basacote	Basacote + YaraMila	YaraMila	Mean A
	Shoot Fresh Weight (g)			
	Shoot dry weight (g)			
100% P	45.07 b	47.80 a	34.11 e	42.32 a
70%P + 30%M	34.80 e	40.20 d	23.51 h	32.83 c
50%P + 50%M	42.20 c	42.80 c	26.35 g	37.11 b
30%P + 70%M	31.02 f	16.11 k	19.17 i	22.10 d
100%M	17.58 j	8.51 l	5.40 m	10.50 e
Mean B	34.13 a	31.09 b	21.71 c	
	Root dry weight (g)			
100% P	13.60 b	12.97 c	11.13 d	12.23 a
70%P + 30%M	11.20 d	14.10 a	8.15 h	11.15 b
50%P + 50%M	14.08 a	14.04 a	9.00 g	12.37 a
30%P + 70%M	10.32 e	13.90 ab	5.04 i	9.75 c
100%M	8.02 h	9.86 f	4.09 j	7.32 d
Mean B	11.44 b	12.77 a	7.48 c	

(A) composed of various combinations of peatmoss (P) and miscanthus straw (M), along with one of three fertilizers (B), including either Basacote (15-11-13) or water soluble YaraMila Complex (12-5-15), either alone or in combination. Different lower case letters within mean A, mean B and A × B interaction indicate statistically significant differences at the significance level of (*p* < 0.05) by Fisher’s test.

**Table 8 plants-12-01639-t008:** Chlorophyll contents (mg/g) in leaves of *Sedum spectabile* ‘Stardust’ grown in containers with five different substrates.

Substrate (A)	Fertilization (B)			
	Basacote	Basacote + YaraMila	YaraMila	Mean A
	Chlorophyll a			
100% P	0.174 d	0.220 b	0.164 e	0.186 b
70%P + 30%M	0.112 g	0.133 f	0.080 h	0.109 e
50%P + 50%M	0.110 g	0.169 de	0.104 g	0.128 d
30%P + 70%M	0.132 f	0.194 c	0.173 de	0.166 c
100%M	0.531 a	0.167 de	0.088 h	0.262 a
Mean B	0.212 a	0.176 b	0.122 c	
	Chlorophyll b			
100% P	0.148 a	0.112 c	0.107 cd	0.122 a
70%P + 30%M	0.087 fg	0.090 f	0.070 i	0.082 b
50%P + 50%M	0.06 4i	0.101 e	0.083 g	0.083 b
30%P + 70%M	0.102 de	0.129 b	0.131 b	0.121 a
100%M	0.058 j	0.110 c	0.076 h	0.081 b
Mean B	0.092 b	0.109 a	0.093 b	
	Total chlorophyll			
100% P	0.322 b	0.331 b	0.271 d	0.308 b
70%P + 30%M	0.200 f	0.224 e	0.150 h	0.191 e
50%P + 50%M	0.174 g	0.271 d	0.187 f	0.210 d
30%P + 70%M	0.233 e	0.324 b	0.304 c	0.287 c
100%M	0.589 a	0.277 d	0.165 g	0.343 a
Mean B	0.304 a	0.285 b	0.215 c	

(A) composed of various combination of peatmoss (P) and miscanthus straw (M), along with one of three fertilizers (B) including either Basacote (15-11-13) or water soluble YaraMila Complex (12-5-15) either alone or in combination. Different lower case letters within mean A, mean B and A × B interaction indicate statistically significant differences at the significance level (*p* < 0.05) by Fisher’s test.

**Table 9 plants-12-01639-t009:** Chlorophyll contents (mg/g) in leaves of *Hydrangea arborescence* ‘Annabelle’ grown in containers with five different substrates.

Substrate (A)	Fertilization (B)			
	Basacote	Basacote + YaraMila	YaraMila	Mean A
	Chlorophyll a			
100% P	0.500 h	0.936 a	0.907 b	0.781 a
70%P + 30%M	0.438 i	0.604 g	0.502 h	0.515 d
50%P + 50%M	0.619 g	0.723 e	0.690 f	0.678 c
30%P + 70%M	0.496 h	0.806 d	0.872 c	0.725 b
100%M	0.518 h	0.623 g	0.377 j	0.506 d
Mean B	0.514 c	0.739 a	0.670 b	
	Chlorophyll b			
100% P	0.359 f	0.504 a	0.469 b	0.444 a
70%P + 30%M	0.206 l	0.311 h	0.272 j	0.263 d
50%P + 50%M	0.304h i	0.388 e	0.355 f	0.349 c
30%P + 70%M	0.297 i	0.456 c	0.443 d	0.399 b
100%M	0.249 k	0.334 g	0.205 l	0.263 d
Mean B	0.283 c	0.399 a	0.349 b	
	Total chlorophyll			
100% P	0.859 i	1.440 a	1.376 b	1.225 a
70%P + 30%M	0.644 k	0.915 h	0.774 j	0.778 d
50%P + 50%M	0.923 h	1.111 e	1.045 f	1.027 c
30%P + 70%M	0.793 j	1.263 d	1.315 c	1.124 b
100%M	0.767 j	0.958 g	0.582 l	0.769 d
Mean B	0.797 c	1.137 a	1.019 b	

(A) composed of various combinations of peatmoss (P) and miscanthus straw (M), along with one of three fertilizers (B), including either Basacote (15-11-13) or water soluble YaraMila Complex (12-5-15), either alone or in combination. Different lower case letters within mean A, mean B and A × B interaction indicate statistically significant differences at the significance level of (*p* < 0.05) by Fisher’s test.

**Table 10 plants-12-01639-t010:** Foliar nutrient contents (mg/g) in *Sedum spectabile* ‘Stardust’ grown in containers with five different substrates.

Substrate (A)	Fertilization (B)			
	Basacote	Basacote + YaraMila	YaraMila	Mean A
	NO_3−_			
100% P	21.7 cd	29.3 a	22.3 c	24.4 a
70%P + 30%M	20.3 def	18.7 f	20.7 cde	19.9 c
50%P + 50%M	24.3 b	20.3 def	19.7 ef	21.4 b
30%P + 70%M	19.3 ef	16.3 g	16.7 g	17.4 d
100%M	16.3 g	137 h	14.3 h	14.8 e
Mean B	20.4 a	19.7 a	18.7 b	
	P			
100% P	182 i	247 f	372 a	267 a
70%P + 30%M	198 h	227 g	300 c	242 d
50%P + 50%M	277 e	287 d	203 h	255 b
30%P + 70%M	197 h	347 b	197 h	247 c
100%M	201 h	226 g	275 e	234 e
Mean B	211 b	267 a	269 a	
	K			
100% P	1817 ef	1467 h	2017 c	1767 c
70%P + 30%M	1617 g	1792 f	2642 a	2016 a
50%P + 50%M	1867 de	1900 d	2183 b	1983 b
30%P + 70%M	1142 i	1908 d	1933 d	1661 d
100%M	1192 i	1483 h	1492 h	1389 e
Mean B	1527 c	1710 b	2053 a	
	Ca			
100% P	6208 b	4067 i	5683 d	5319 c
70%P + 30%M	6117 b	4933 f	5933 c	5661 b
50%P + 50%M	7792 a	5108 e	4758 g	5886 a
30%P + 70%M	5742 d	3942 j	4367 h	4683 d
100%M	4767 g	3442 l	3642 k	3950 e
Mean B	6125 a	4298 c	4877 b	
	Mg			
100% P	146 e	204 a	199 a	183 a
70%P + 30%M	153 e	180 b	171 c	168 c
50%P + 50%M	201 a	198 a	161 d	187 a
30%P + 70%M	150 e	150 e	128 f	143 d
100%M	145 e	205 a	168 cd	173 b
Mean B	159 c	187 a	166 b	

(A) composed of various combinations of peatmoss (P) and miscanthus straw (M), along with one of three fertilizers (B), including either Basacote (15-11-13) or water soluble YaraMila Complex (12-5-15), either alone or in combination. Different lower case letters within mean A, mean B and A × B interaction indicate statistically significant differences at the significance level of (*p* < 0.05) by Fisher’s test.

**Table 11 plants-12-01639-t011:** Foliar nutrient contents (mg/g) in *Hydrangea arborescence* ‘Annabelle’ grown in containers with five different substrates.

Substrate (A)	Fertilization (B)			
	Basacote	Basacote + YaraMila	YaraMila	Mean A
	NO_3−_			
100% P	14.1 a	13.1 c	12.9 c	13.4 a
70%P + 30%M	13.5 b	11.3 e	12.1 d	12.3 b
50%P + 50%M	13.0 c	10.9 f	10.8 f	11.6 c
30%P + 70%M	11.3 e	10.6 f	10.6 f	10.8 d
100%M	10.6 f	10.6 f	10.8 f	10.6 e
Mean B	12.5 a	11.3 b	11.4 b	
	P			
100% P	142 j	344 a	220 f	235 b
70%P + 30%M	179 g	167 h	231 e	192 d
50%P + 50%M	156 i	230 e	180 g	189 d
30%P + 70%M	288 c	131 k	214 f	211 c
100%M	305 b	274 d	301 b	293 a
Mean B	214 b	229 a	229 a	
	K			
100% P	83 gh	157 c	103 e	114 b
70%P + 30%M	87 fg	173 b	83 gh	114 b
50%P + 50%M	120 d	247 a	87 fg	151 a
30%P + 70%M	87 fg	127 d	73 hi	96 c
100%M	63 ij	97 ef	53 j	71 d
Mean B	88 b	160 a	80 c	
	Ca			
100% P	1550 de	1337 gh	1570 cd	1486 b
70%P + 30%M	1350 fgh	1397 f	1300 h	1349 c
50%P + 50%M	2160 a	1607 c	1983 b	1917 a
30%P + 70%M	1503 e	1383 fg	1587 cd	1491 b
100%M	593 i	547 i	583 i	574 d
Mean B	1431 a	1254 c	1405 b	
	Mg			
100% P	282 d	199 j	227 h	236 d
70%P + 30%M	290 d	395 a	283 d	322 a
50%P + 50%M	214 i	340 b	252 ef	268 b
30%P + 70%M	247 efg	242 fg	253 e	247 c
100%M	305 c	254 e	239 g	266 b
Mean B	268 b	286 a	251 c	

(A) composed of various combinations of peatmoss (P) and miscanthus straw (M), along with one of three fertilizers (B), including either Basacote (15-11-13) or water soluble YaraMila Complex (12-5-15), either alone or in combination. Different lower case letters within mean A, mean B and A × B interaction indicate statistically significant differences at the significance level of (*p* < 0.05) by Fisher’s test.

**Table 12 plants-12-01639-t012:** pH and EC of substrates in *Sedum spectabile* ‘Stardust’ grown in containers with five different substrates.

Substrate (A)	Fertilization (B)			
	Basacote	Basacote + YaraMila	YaraMila	Mean A
	pH			
100% P	6.5 ab	5.3 h	5.7 fg	5.8 d
70%P + 30%M	5.6 g	5.7 fg	5.8 f	5.7 d
50%P + 50%M	6.0 de	6.1 d	6.4 bc	6.1 c
30%P + 70%M	6.4 bc	6.2 bc	6.5 ab	6.4 b
100%M	6.6 a	6.3 cd	6.6 a	6.5 a
Mean B	6.2 a	5.9 b	6.2 a	
	EC (mS/cm)			
100% P	893 bc	1447 a	645 g	995 a
70%P + 30%M	823 d	888 c	623 g	778 c
50%P + 50%M	691 f	780 e	910 b	794 b
30%P + 70%M	430 i	671 f	376 j	493 d
100%M	302 k	534 h	369 j	402 e
Mean B	628 b	864 a	585 c	

(A) composed of various combinations of peatmoss (P) and miscanthus straw (M), along with one of three fertilizers (B), including either Basacote (15-11-13) or water soluble YaraMila Complex (12-5-15), either alone or in combination. Different lower case letters within mean A, mean B and A × B interaction indicate statistically significant differences at the significance level of (*p* < 0.05) by Fisher’s test.

**Table 13 plants-12-01639-t013:** pH and EC of substrates in *Hydrangea arborescence* ‘Annabelle’ grown in containers with five different substrates.

Substrate (A)	Fertilization (B)			
	Basacote	Basacote + YaraMila	YaraMila	Mean A
	pH			
100% P	8.4 a	7.9 b	7.7 bc	8.0 a
70%P + 30%M	7.7 bc	7.4 de	7.5 cd	7.5 b
50%P + 50%M	7.2 ef	7.1 f	7.3	7.2 c
30%P + 70%M	7.4 de	7.3 ef	7.3 ef	7.3 c
100%M	7.4 de	7.1 f	7.4 de	7.3 c
Mean B	7.6 a	7.4 b	7.5 b	
	EC (mS/cm)			
100% P	605 e	742 c	669d	672 a
70%P + 30%M	493 f	796 a	430h	572 c
50%P + 50%M	751 b	795 a	440g	662 b
30%P + 70%M	353 j	411 i	337k	367 d
100%M	191 l	339 k	168m	233 e
Mean B	478 b	617 a	408 c	

(A) composed of various combinations of peatmoss (P) and miscanthus straw (M), along with one of three fertilizers (B), including either Basacote (15-11-13) or water soluble YaraMila Complex (12-5-15), either alone or in combination. Different lower case letters within mean A, mean B and A × B interaction indicate statistically significant differences at the significance level of (*p* < 0.05) by Fisher’s test.

**Table 14 plants-12-01639-t014:** Substrate nutrient contents in *Sedum spectabile* ‘Stardust’ grown in containers with five different substrates.

Substrate (A)	Fertilization (B)			
	Basacote	Basacote + YaraMila	YaraMila	Mean A
	N Total (% d.w.)			
100% P	1.59 e	1.86 ab	1.75 cd	1.73 a
70%P + 30%M	1.51 e	1.97 a	1.85 bc	1.78 a
50%P + 50%M	1.60 e	1.79 bcd	1.56 e	1.65 b
30%P + 70%M	1.52 e	1.74 d	1.72 d	1.66 b
100%M	1.50 e	1.39 f	1.60 e	1.50 c
Mean B	1.54 c	1.75 a	1.70 b	
	NO_3−_ (mg/dm^3^)			
100% P	17.0 efg	10.7 h	18.3 e	15.3 d
70%P + 30%M	15.3 g	46.3 a	17.3 ef	26.3 a
50%P + 50%M	22.3 bc	20.3 d	22.7 b	21.8 b
30%P + 70%M	16.3 fg	20.3 d	20.7 cd	19.1 c
100%M	5.7 i	4.3 i	5.3 i	5.1 d
Mean B	15.3 c	20.4 a	16.9 b	
	P (mg/dm^3^)			
100% P	23 k	116 e	66 h	69 d
70%P + 30%M	39 j	100 f	228 a	122 a
50%P + 50%M	17 l	127 c	88 g	77 c
30%P + 70%M	23 k	123 d	86 g	77 c
100%M	53 i	118 e	162 b	111 b
Mean B	31 c	117 b	126 a	
	K (mg/dm^3^)			
100% P	57 e	183 a	53 ef	98 a
70%P + 30%M	53 ef	123 b	83 d	87 b
50%P + 50%M	57 e	117 b	50 ef	74 c
30%P + 70%M	43 f	97 c	27 g	56 e
100%M	93 cd	57 e	53 ef	68 d
Mean B	61 b	115 a	53 c	
	Ca (mg/dm^3^)			
100% P	1193 d	997 h	1123 f	1104 c
70%P + 30%M	1197 d	963 i	1083 g	1081 d
50%P + 50%M	1223 c	1356 b	1473 a	1351 a
30%P + 70%M	1160 e	1227 c	1167 e	1184 b
100%M	483 j	343 l	403 k	410 e
Mean B	1051 a	977 b	1050 a	
	Mg (mg/dm^3^)			
100% P	119 g	128 f	163 b	136 a
70%P + 30%M	116 g	150 c	145 d	137 a
50%P + 50%M	117 g	129 f	168 a	138 a
30%P + 70%M	138 e	143d e	129 f	137 a
100%M	63 i	65 i	72 h	66 b
Mean B	111 c	123 b	135 a	

(A) composed of various combinations of peatmoss (P) and miscanthus straw (M), along with one of three fertilizers (B), including either Basacote (15-11-13) or water soluble YaraMila Complex (12-5-15), either alone or in combination. Different lower case letters within mean A, mean B and A × B interaction indicate statistically significant differences at the significance level of (*p* < 0.05) by Fisher’s test.

**Table 15 plants-12-01639-t015:** Substrate nutrient contents in *Hydrangea arborescence* ‘Annabelle’ grown in containers with five different substrates.

Substrate (A)	Fertilization (B)			
	Basacote	Basacote + YaraMila	YaraMila	Mean A
	N Total (% d.w.)			
100% P	2.89 cdef	3.60 ab	3.64 ab	3.37 a
70%P + 30%M	2.53 efg	4.05 a	3.48 b	3.35 a
50%P + 50%M	2.89 cdef	3.94 a	2.93 cde	3.25 a
30%P + 70%M	2.86 def	3.33 bc	3.21 bcd	3.13 a
100%M	2.43 fg	2.17 ef	2.19 g	2.45 b
Mean B	2.67 c	3.53 a	3.14 b	
	NO_3−_ (mg/dm^3^)			
100% P	12.2 h	10.6 i	12.6 h	11.8 d
70%P + 30%M	13.9 g	13.8 g	13.4 g	13.7 c
50%P + 50%M	28.7 a	18.0 f	27.1 b	24.6 a
30%P + 70%M	19.2 e	23.4 d	24.8 c	22.5 b
100%M	6.7 k	7.4 j	7.8 j	7.3 e
Mean B	16.1 b	14.6 c	17.2 a	
	P (mg/dm^3^)			
100% P	27 k	45 i	87 e	53 d
70%P + 30%M	14 l	73g	54 h	47 e
50%P + 50%M	28 k	107 c	104 c	79 b
30%P + 70%M	25 k	79 f	100 d	68 c
100%M	42 j	163 a	144 b	116 a
Mean B	27 c	94 b	98 a	
	K (mg/dm^3^)			
100% P	2633 f	3133 b	1317 l	2361 c
70%P + 30%M	2708 e	2767 d	3383 a	2953 a
50%P + 50%M	2508 g	2617 f	2833 c	2653 b
30%P + 70%M	2133 h	2133 h	2508 g	2258 d
100%M	1617 k	1816 j	1908 i	1781 e
Mean B	2320 c	2493 a	2390 b	
	Ca (mg/dm^3^)			
100% P	3025 a	2100 de	1917f	2347 a
70%P + 30%M	2675 b	1758 g	1525h	1986 c
50%P + 50%M	2800 b	1525 h	1525 h	1950 cd
30%P + 70%M	2325 c	1867 fg	1525 h	1905 d
100%M	2200 cd	2050 e	1975 ef	2075 b
Mean B	2605 a	1860 b	1693 c	
	Mg (mg/dm^3^)			
100% P	117 g	124 f	145 c	129 c
70%P + 30%M	116 g	141 cd	132 e	130 c
50%P + 50%M	137 d	165 a	153 b	151 a
30%P + 70%M	121 fg	166 a	117 g	134 b
100%M	39 j	68 h	46 i	51 d
Mean B	106 c	133 a	119 b	

(A) composed of various combinations of peatmoss (P) and miscanthus straw (M), along with one of three fertilizers (B), including either Basacote (15-11-13) or water soluble YaraMila Complex (12-5-15), either alone or in combination. Different lower case letters within mean A, mean B and A × B interaction indicate statistically significant differences at the significance level of (*p* < 0.05) by Fisher’s test.

## Data Availability

The data presented in this study are available in the article.

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
