# Peer review of "Assessment of Fresh Miscanthus Straw as Growing Media Amendment in Nursery Production of Sedum spectabile ‘Stardust’ and Hydrangea arborescens ‘Annabelle’"

_plants, 2023, doi:10.3390/plants12081639_

Round 1

Reviewer 1 Report

 This study tried to to assess the influence of fresh miscanthus straw shreds as a component of growing media in nursery production of perennial Sedum spectabile 'Stardust' and woody shrub Hydrangea arborescens ‘Annabelle’, which possesses some merits. Basically, both topic and subject fall within the general scope of the Journal, and the manuscript is somewhat straightforward and understandable. Indeed, lots of data have been collected, and the analysis seems to be performed to a good technical standard. However, at the present form, it is hard for me to judge whether the conclusions are fully supported by the data. Meanwhile, some improvements are needed in manuscript organizing, language editing and conclusions. Therefore, I recommend it would be reconsidered for publication after major revision.

Specific Comments:

1) In the Introduction Section, the authors have not explained why they select perennial Sedum spectabile 'Stardust' and woody shrub Hydrangea arborescens 'Annabelle as research materials. 

2) When evaluating the effects of treatments on plant performance of height and diameter, I do think net growth is more better than total mean growth.

3) Too much data were presented in the manuscript, I do think some data should be presented as supplementary materials.

4) You measured lots of data and presented in the result section, but no comprehensive assessments were conducted to evaluate the treatment effects, which is the big drawback.

5) Also, I suggest the Discussion should be seperated from the Result section.

Author Response

Dear Reviewer,

according to the suggestion, discussion was extended and a few additional literatures were cited. Some statements of common knowledge were left without citations.

Methods were supported with protocol references and some more details were added to this section.

Sincerely,

Przemysław Bąbelewski

Reviewer 2 Report

Dear Authors

The manuscript is well documented, with sufficient results. The design and outcome is clear and the posed question has been answered.

The only thing I would suggest would be to push a bit further the discussion and increase the amount of references. Given that the introduction has 34 references it is a pity that the rest of the manuscript only adds another 17.

Also, material and methods could be written in more details, and maybe with the addition of some references towards the analysis protocol,

Author Response

Dear Reviewer,

please, find response to your comments below:

  • As suggested, explanation of reasoning behind selecting Sedum and Hydrangea was added.
  • Mean growth seems to be reasonable in the case of this study, where the main evaluation was trying to answer the question about final quality of plants deciding about their market value.
  • Following the suggestion, some of the tables were removed and some of them were shortened by removing data showing the same trends. Data that were removed were kept only as statements in reference to existing data. Authors are willing to move other tables to supplementary materials upon precise reviewer’s request.
  • Sections Results and Discussion and Conclusions were expanded and modified to provide more comprehensive assessment of this study.
  • As many data show clear tendencies and can be described in very concise way, it makes more sense to keep these two chapters together, which is also a format accepted by this journal.

Sincerely,

Przemysław Bąbelewski

Round 2

Reviewer 1 Report

The authors have made the efforts to improve the manuscript, thus I suggest accepting the paper for publishing though there are some shortages in the  organization of the paper and the description of the conclusions. 

Author Response

The manuscript has been revised upon the academic editor request.